# In Vivo Rodent Models of Type 2 Diabetes and Their Usefulness for Evaluating Flavonoid Bioactivity

**DOI:** 10.3390/nu11030530

**Published:** 2019-02-28

**Authors:** Jia-You Fang, Chih-Hung Lin, Tse-Hung Huang, Shih-Yi Chuang

**Affiliations:** 1Pharmaceutics Laboratory, Graduate Institute of Natural Products, Chang Gung University, Kweishan, Taoyuan 33302, Taiwan; fajy@mail.cgu.edu.tw; 2Chinese Herbal Medicine Research Team, Healthy Aging Research Center, Chang Gung University, Kweishan, Taoyuan 33302, Taiwan; 3Research Center for Food and Cosmetic Safety and Research Center for Chinese Herbal Medicine, Chang Gung University of Science and Technology, Kweishan, Taoyuan 33302, Taiwan; 4Department of Anesthesiology, Chang Gung Memorial Hospital, Kweishan, Taoyuan 33302, Taiwan; 5Center for General Education, Chang Gung University of Science and Technology, Kweishan, Taoyuan 33302, Taiwan; chlin@mail.cgust.edu.tw; 6Department of Traditional Chinese Medicine, Chang Gung Memorial Hospital, Keelung 20401, Taiwan; huangtsehung@gmail.com; 7School of Traditional Chinese Medicine, Chang Gung University, Kweishan, Taoyuan 33302, Taiwan; 8Graduate Institute of Health Industry Technology, Chang Gung University of Science and Technology, Kweishan, Taoyuan 33303, Taiwan; 9School of Nursing, National Taipei University of Nursing and Health Sciences, Taipei 112, Taiwan

**Keywords:** flavonoid, antioxidant, type 2 diabetes, animal model

## Abstract

About 40% of the world’s population is overweight or obese and exist at risk of developing type 2 diabetes mellitus (T2D). Obesity is a leading pathogenic factor for developing insulin resistance (IR). It is well established that IR and a progressive decline in functional β-cell mass are hallmarks of developing T2D. In order to mitigate the global prevalence of T2D, we must carefully select the appropriate animal models to explore the cellular and molecular mechanisms of T2D, and to optimize novel therapeutics for their safe use in humans. Flavonoids, a group of polyphenols, have drawn great interest for their various health benefits, and have been identified in naturally occurring anti-diabetic compounds. Results from many clinical and animal studies demonstrate that dietary intake of flavonoids might prove helpful in preventing T2D. In this review, we discuss the currently available rodent animal models of T2D and analyze the advantages, the limitations of each T2D model, and highlight the potential anti-diabetic effects of flavonoids as well as the mechanisms of their actions.

## 1. Type 2 Diabetes

Diabetes mellitus is a chronic metabolic disease that is characterized by a relative lack of insulin, resulting in hyperglycemia. Chronic hyperglycemia can lead to a variety of complications such as neuropathy, nephropathy, and retinopathy, as well as increased risk of cardiovascular disease [1]. The prevalence of diabetes mellitus is rapidly rising. The International Diabetes Federation (IDF) estimated that 425 million adults (aged 20–79 years) had diabetes mellitus globally in 2017, and further estimated that a US $727 billion (12% of global health expenditure) was spent on treating diabetes mellitus and its related complications [2]. About 1 in 11 adults have diabetes mellitus, with 90% of these adults having type 2 diabetes mellitus (T2D); Asia is the center of this global T2D epidemic [3]. It is now well-recognized that T1D is an autoimmune disorder characterized by the destruction of insulin-producing pancreatic β-cells [4]. However, T2D is characterized by insulin resistance (IR) and the pancreatic β-cell failure to sufficiently compensate (Figure 1). T2D is an acquired syndrome characterized by several defects in the regulation of glucose homeostasis, including elevated blood glucoside levels, increased hepatic glucose production, deficient insulin secretion, IR, and/or pancreas β-cell dysfunction [1,2]. As reported by the World Health Organization [3], T2D comprises approximately 90% of all cases of diabetes, and an estimated 15 million people globally have T2D, a figure that could double by 2025. While the genetic architecture might partially determine an individual’s response to environmental changes, the main drivers of the global epidemic of T2D are the rise in obesity, a sedentary lifestyle, energy-dense diets, and population ageing. Therefore, animal models of T2D tend to include models of IR and/or models of β-cell failure. Many animal models of T2D are obese, reflecting the human condition where obesity is closely linked to T2D development. Preclinical animal models have provided considerable valuable information about obesity and T2D. Despite constant improvements and refinements in cell-based applications, careful metabolic assessment of compound effects in in vivo models is vital before drugs can be considered for clinical evaluation and commercialization.

## 2. Animal Models of T2D

Animal models remain indispensable for discovering, validating, and optimizing novel therapeutics for their safe use in humans. Different T2D animal models, ranging from non-mammalian models to nonhuman primates, each have distinct advantages and limitations. Here, we have summarized the key information on currently available animal models of T2D, offering guidance as to the usefulness, advantages and limitations of these models depending on the species (Figure 2), altered pathway, environmental conditions, and genetic background.

### 2.1. Non-Mammalian Models

As a general rule of the laboratory T2D study, roundworm *C. elegans* [5], fruit fly *Drosophila melanogaster* [6], and zebrafish *Danio rerio* [7] are the non-mammalian models usually established in laboratory settings. Schlotterer et al. [8] established *C. elegans* as a model for diabetes research through the responses of *C. elegans* to being fed high glucose concentrations; it was shown that their lifespan is reduced by increased reactive oxygen species (ROS) generation and advanced glycation end products (AGEs)-modification of proteins. Moreover, the inhibition of the activities of the DAF-16 and the heat shock factor (HSF-1), which are also both inhibited by insulin signaling, were also suggested to underlie glucose-induced lifespan reductions [9]. *Drosophila melanogaster* could emerge as a powerful system for dissecting the genetics of IR and secretion because the mechanisms of glucose homeostasis are conserved between flies and humans, and the fruit fly allows for substantial ease of experimental and genetic manipulation in comparison to rodent models [10]. Park et al. [11] generated a double-tagged insulin-like peptide 2 (ilp2HF) to monitor its secretion. They found a marked increase in ilp2HF-circulating levels upon re-feeding after a 24 h fast; it is likely a result of glucose sensing by glucose transporter type (GLUT) 1 in the insulin-producing cells (IPCs), as IPC-specific knockdown of *Glut1* decreased circulating ilp2HF. Insulin-resistant *Drosophila* have also been generated by rearing flies on high-sugar diet (HSD). HSD causes IR with decreasing insulin-like peptides expression; these flies develop hyperglycemia through the production of a robust suppression of Lst, a negative regulator of insulin-like peptides production and secretion [12,13]. Both nutritional and genetic approaches have also been used to generate T2D models in zebrafish. Several studies suggest that the immersion of zebrafish in glucose solution is a widely used model to induce diabetic phenotypes, including elevated blood glucose levels and impaired response to exogenous insulin [14,15]. Chen et al. developed two transgenic zebrafish models of IR in skeletal muscle and liver, a result achieved through ablation of the insulin receptors [16,17]. Another type of diabetes, MODY (maturity-onset diabetes of the young), is a rare, autosomal dominant, noninsulin-dependent, and monogenic form of diabetes resulting from pancreatic β-cell dysfunction [18]. A zebrafish mutant line, with mutations in hepatocyte nuclear factor 1β, exhibits pancreas hypoplasia and reduced β-cell numbers [19,20]; it closely mimics established human diseases, such as the MODY form of diabetes. Curado et al. [21] were able to study β-cell regenerative capacity by the addition of a prodrug metronidazole that induces cell death of the β-cells in zebrafish transgenic lines. Non-mammalian models have the advantage of low maintenance cost, short life cycle, and availability of long-term gene-editing tools quantification. However, their translational value is limited given their physiology difference to mammals.

### 2.2. Large Animal Models

Dogs and pigs are the large animal models used for translational studies in research on obesity and diabetes mellitus [22,23]. The canine model is especially powerful in allowing quantification of liver glucose uptake; researchers can induce diabetes mellitus in dogs with pancreatectomy or with the use of alloxan and/or streptozotocin (STZ) [24,25]. In these models, metabolic defects are evident in all of them, including increased visceral, subcutaneous, and total adipose tissue mass, increased IR and a β-cell defect. The dog model also provides invasive measures and assessments impossible in humans or rodent models, particularly for studies involving oral administration of compounds because canine gastrointestinal anatomy and physiology are highly similar to those of human beings. The pig is another large animal model used for diabetes mellitus; it shares the structure and function of the gastrointestinal tract, the development and morphology of the pancreas, and overall metabolic status with humans [26]. Genetic engineering of pigs represents an approach for diabetes mellitus research, including transgenic pigs expressing a dominant-negative glucose-dependent insulinotropic polypeptide receptor (GIPR) [27] and ubiquitous expression of a dominant-negative human hepatocyte nuclear factor 1α (HNF1A) [28]. High fat and cholesterol (HFC) diets with or without STZ have been adapted for the establishment of pig models for T2D [29,30]. While pigs are increasingly used as models for obesity and diabetes mellitus research, dogs remain an important model in this field.

### 2.3. Nonhuman Primate Models

While other animal models provide insights into the mechanisms underlying T2D, they are limited in their translatability to humans. The nonhuman primate shares more metabolic similarities with the human, making it an ideal model for the investigation of T2D and use in preclinical trials [31]. A long history of studying nonhuman primates for translational research in T2D, and the most commonly used species include rhesus macaques (*Macaca mulatta*), cynomolgus monkeys (*Macaca fascicularis*), baboons (*Papio species*), African Green Monkeys (*Chlorocebus species*), and common marmosets (*Callithrix jacchus*) [32]. A number of models are in use for the study of T2D in nonhuman primates, including high fructose-fed nonhuman primate models [33], diet-induced nonhuman primate models [34], and aged-spontaneous T2D nonhuman primate models [35]. The presence of islet amyloidosis, IR to β-cell failure, and overt diabetes mellitus characteristic of T2D in humans is also observed in diabetic monkeys, implicating a similar etiology of islet lesions in monkeys and humans [36,37]. Given the substantial cost of conducting clinical trials with humans, studies in non-human primate models represent a cost-effective avenue for studying T2D strategy. However, the use of nonhuman primate models is an expense in biomedical research, and relatively few laboratories are equipped to accommodate such studies.

### 2.4. Rodent Models

#### 2.4.1. Monogenic Models

The rodent has proven to be a reliable model for discovering and validating new treatments for T2D (Table 1) [38]. Compared to non-mammalian species, as mammals, the physiology of mice and rats is closer to that of humans. We here review commonly used rodent models to study glucose metabolism which show good translatability to the diagnostic of T2D in humans. While obesity in humans is rarely caused by a monogenic mutation, monogenic models of obesity are commonly used in T2D research. The most widely used monogenic models of obesity are defective in leptin signaling. Leptin induces satiety, and thus, a lack of functional leptin in these animals causes hyperphagia and subsequent obesity. These models include the Lep^ob/ob^ mouse [39,40,41], which is deficient in leptin, and the Lepr^db/db^ mouse and Zucker Diabetic Fatty (ZDF) rat, which are deficient in the leptin receptor [42,43]. These models are also used to test new therapies for T2D.

The Lep^ob/ob^ mouse is a model of severe obesity derived from a spontaneous mutation on chromosome 6 discovered in an outbred colony at Jackson Laboratory in 1949 [44]. The phenotype was bred into C57BL/6J mouse, but it was not until 1994 that the mutated protein was identified as leptin [45]. The obesity was observed at 2 weeks of age, and the mice develop hyperinsulinemia. By 7–12 weeks, hyperglycemia is apparent; at this point, pancreatic β-cell compensation occurs, and increased insulin levels bring glucose homeostasis under control, after which they fall as the mouse ages. In addition, other metabolic aberrations include hyperlipidemia; hepatic fatty acid synthesis was increased 6-fold per total liver and 2.2-fold per total small intestine [46]. The severity of the diabetes in Lep^ob/ob^ mouse is strongly dependent upon the strain’s genetic background. With the C57BL/KsJ background, Lep^ob/ob^ mouse develops more severe hyperglycemia and diabetes, accompanied by increasing insulin levels, followed by β-cell failure, and often dies by 6 months of age [47]. Other different background strains have been studied, including the BALB/cJ background [48] at 27 weeks of age have a 35-40% reduction in weight gain with elevations in fed insulin and plasma triglycerides levels. The Lepr^db/db^ mouse originated from the Jackson Laboratory, and are deficient in the leptin receptor [49]. The Lepr^db/db^ resembles Lep^ob/ob^ mouse in terms of a rapid development of obesity, but the diabetes syndrome is more severe. Depending upon their genetic background, the Lepr^db/db^ mouse in the C57BL/6J inbred strain is a rapid development of obesity at 3–4 weeks, blood glucose level at 7 weeks of age and hyperglycemia developing at 4-8 weeks. The Lepr^db/db^ mouse with the C57BL/KsJ background develops more severe diabetes and has a relatively short lifespan [50].

The Zucker Fatty (ZF) rats were discovered in 1961 after a cross of Merck M-strain and Sherman rats. They harbor a missense mutation (fatty, fa) in the leptin receptor gene (Lepr) and become obese at around 4 weeks of age [51]. These rats are also hyperinsulinemic, hyperlipidemic, and hypertensive. ZDF rats derived from a mutation in the ZF rat strains exhibit less obesity than the ZF rats but have more severe IR, which they are unable to compensate for due to increased apoptosis levels in β-cells; they are widely used for research on T2D [52]. They are characterized by initial hyperinsulinemia at 8 weeks of age followed by decreased insulin levels and signs of diabetic complications usually develop at 10 weeks [50,53].

Those findings underline the fact that although the monogenetic models are indeed a valuable and useful animal model of T2D and do have a role in terms of teasing out the mechanism and mode of action of these diseases, but the human obesity phenotype is caused by the interplay between a long list of genes and the environment. Notably, few cases of human obesity can be accounted for mutations in leptin or the leptin receptor. Thus, it does not reflect the complete background of T2D in humans based on its monogenic cause of obesity or T2D and will therefore not always be predictive of the effects of pharmaceutical treatments in humans.

#### 2.4.2. Polygenic Models

Polygenic models of obesity may provide a more accurate model of the human condition [54]. A variety of different polygenic mouse models of obesity, glucose intolerance and diabetes exist, allowing a variety of genotypes and susceptibilities to be studied. However, unlike the monogenic models, there are no wild-type controls. Since the obesity is induced by environmental manipulation rather than genes, it is thought to model the human situation more accurately than monogenic models. The diet-induced obesity (DIO) model has considerable direct validity with human obesity, and is a widely used paradigm to study the interaction of diet and genes in manifest obesity and IR [55]. The inbred C57BL/6J mouse strain, first described in 1988, is a polygenic obesity-prone mouse strain widely used as a model for DIO [56] because it is prone to develop obesity and glucose intolerance and moderate IR, making them especially intriguing models to study human obesity resistance. However, a report [57] on the effects of a drug on overt diabetes end points, suggests that the C57BL/6J model for DIO is not the ideal choice because this strain rarely develops frank hyperglycemia and/or islet atrophy when fed an obesogenic diet. In contrast, the rather closely related but distinct C57BL/6N strain develops hepatosteatosis, hyperglycaemia and hyperinsulinaemia following 3 weeks on an HFD [58,59]. It means that they have considerable genetic variation, which investigators should take into account when determining sample size. The DIO rat and mouse offer more human-like models, where the obesity is based on several factors, including an excess intake of calories.

The New Zealand Obese (NZO) mouse is another inbred polygenic strain created by selective breeding that develops obesity and T2D. Adiposity in the NZO mouse is driven by a moderate hyperphagia, which may be a consequence of leptin resistance as these mice are hyperleptinemic at 9–12 weeks of age, and causes disturbances related to both pancreatic and hepatic defects [60]. When fed a carbohydrate-containing diet, there is a dramatic effect on the pancreatic islets including a loss of serine/threonine-protein kinase (AKT) activation, decreased expression of GLUT2 essential for insulin synthesis and β-cell integrity [61]. Similar to what occurs in humans, the onset of T2D in NZO mice decidedly depends on the degree of hepatosteatosis early in life [62]. TALLYHO/Jng mice are another emerging, polygenic model for moderate obesity and diabetes [63]. TALLYHO/Jng mice were inbred from two Theiler Original outbred mice that showed polyuria and glucosuria [64], selecting for mice at each generation that demonstrated β-cell hypertrophy, hyperplasia with hyperinsulinemia, severe dyslipidemia, and hyperglycemia. IR in white adipose tissue is associated with reduced GLUT4 cycling and increased insulin receptor substrate 1 (IRS-1) degradation [65]. Both male and female TALLYHO/Jng mice become moderately obese; hyperglycaemia is limited to male mice, and develops as early as at 10–14 weeks of age [66]. In 1952, Kondo et al. [67] established many mouse strains, and then Nakamura et al. found that the KK mouse among these inbred strains spontaneously develops diabetic characteristics, including obesity, hyperinsulinemia, and IR in both muscle and adipose tissue [68]. This mouse strain also shows signs of diabetic nephropathy [69]. Due to the relatively mild diabetes and obesity in KK mouse, a derivative of this strain is the KK-A^y^ mouse, which was created by transferring the yellow obese gene (A^y^) into KK mouse by crossing yellow obese mice with KK mice [70].

### 2.5. Chemical-Induced Model of T2D

T2D is increasing in prevalence worldwide, and is strongly associated with obesity and IR, as well as defects in pancreatic β-cell function and mass, thus precipitating a disease characterized by impeding the critical regulatory influence of insulin on glucose, lipid and protein metabolism [71]. Characterized and clinically relevant T2D animal models are required to create and achieve the aim of testing new therapeutics. An example of a chemical-induced animal model of diabetes is the high-fat diet/streptozotocin-treated (HFD/STZ) animal model. This model involves a combination of an HFD to bring about hyperinsulinemia, IR, and/or glucose intolerance followed by subsequent injection of a low dose (~30–40 mg/kg intraperitoneally) STZ, which results in severe reduction in functional β-cell mass [72]. Together, these two stressors are designed to mimic the pathology of T2D, though on a shorter timescale than found in the human condition. The key advantage of this non genetic model is that researchers can customize it to resemble the slow pathogenesis of T2D that occurs in most humans, encompassing the slow development from adult-onset DIO to glucose intolerance, IR, the resulting compensatory insulin release, and finally STZ-induced partial β-cell death. Despite its limitations and the wide variety of both the high-fat fed regimen [73] and the STZ treatment [74], the HFD/STZ is a reasonable animal model of T2D mainly representing the later stage of the disease, depending on the amount of residual β-cell mass.

## 3. Flavonoids and Their Effects on T2D

Flavonoids belong to a large group of polyphenols; they mainly accumulate in the edible parts of plants, particularly in fruits, vegetables, herbs, chocolate, tea, seeds, flowers, and red wine [75]. More than 9000 flavonoids have been identified from plant sources [76], and their daily intake varies between 20 mg and 500 mg, mainly from dietary supplements including tea, red wine, apples, onions, and tomatoes [77]. The basic chemical structural of flavonoids consists two benzene rings (A and B) linked by a three-carbon chains, forming an oxygenated heterocycle (ring C); they provide attractive color pigments such as yellow, red, blue, and purple in plants. Flavonoids are classified according to their chemical structure. These medicinal herbs have been traditionally used for the treatment of T2D; among the phytochemical compounds, flavonoids, and their derivatives are more under attention due to their hypoglycemic activity [78]. Flavonoids are classified subclasses based on chemical structures, six of which are: Anthocyanidins, flavan-3-ols, flavonols, flavones, flavanones, and isoflavones (Table 2). Flavonoids have antioxidative properties that protect the body against the deleterious effects of hyperglycemia in T2D, by acting on biological targets such as α-glucosidase, glucose co-transporter, or aldose reductase. These antioxidants have been proposed as potential anti-diabetic drugs by acting as biological targets involved in T2D development. In current years, various approaches have been made to utilize the flavonoids in vivo rodent models by incorporating a few novel methods to improve its antidiabetic activity. They are categorized in Table 3.

### 3.1. Flavonol

Flavonols are the most abundant flavonoids in the plant kingdom. The main dietetic flavonols include quercetin, rutin, kaempferol, isorhamnetin, and myricetin [79,80], while the most ubiquitous compounds are glycosylated derivatives of quercetin and kaempferol; in nature, these two molecules have respectively about 280~350 different glycosidic combinations. Quercetin is ubiquitously present in fruits and vegetables and considered a strong antioxidant, anti-inflammation against proinflammatory cytokines production [81,82], platelet aggregation prevention, and anti-diabetes effects [83,84]. Recently, quercetin has been shown to enhance insulin secretion via Ca^2+^ and ERK1/2 signaling pathway [85], acts as partial agonists of PPARγ [86], and potentiates ERK1/2 phosphorylation [87] in vitro. Studies have been conducted on the effects of quercetin in animals with T2D, quercetin ameliorates hyperglycemia (lower glucose plasma levels) and dyslipidemia (an increase in plasma adiponectin and HDL-cholesterol, decreases in plasma total cholesterol and plasma triacylglycerols) and improves the antioxidant status in type 2 diabetic Lepr^db/db^ mouse [88]. A study showed that transforming the growth factor-β1 (TGF-β1) and connective tissue growth factor (CTGF) has an essential impact on diabetic nephropathy. A report showed that HFD/STZ-induced rats treated with quercetin saw a reduction in their weight ratio of kidney and body, and attenuated expressions of CTGF and TGF-β1 in the renal tissues [89,90]. A recent report indicates that diets rich in flavonoids are associated with a lower incidence of T2D [91]. Among the flavonol subclass, quercetin, and myricetin intake was associated with a lower incidence of T2D in European populations [92]. Another report suggests that daily intakes of quercetin from apple and orange is inversely related to the prevalence of T2D: 8.35% in men and 4.68% in women in a Chinese population [78]. Rutin can be broadly extracted from natural plant sources such as buckwheat, citrus fruits, grapes, lemons, and berries, and was also reported to have anti-obesity and anti-diabetic functions [93]. Rutin potentiates insulin receptor kinase to enhance insulin-dependent GLUT4 translocation through the enhancement of insulin receptor kinase activity, thereby activating the insulin signaling pathway, causing increased GLUT4 translocation and increased glucose uptake [94]. Rutin was also found to activate liver enzymes linked with the gluconeogenic and lipid metabolic processes. It reduced the alanine aminotransferase, aspartate aminotransferase activities and advanced glycation end products level in serum. It also potentiated the phosphorylation of phosphatidylinositol-4,5-bisphosphate 3-kinase (PI3K), Akt, glycogen synthase kinase-3 beta protein in the liver tissue of Lepr^db/db^ mouse [95], and was shown to influence glucose uptake in the rat soleus muscle [96]. Rutin was also reported to reduce the levels of plasma glucose, glycosylated hemoglobin (HbA1c), a glycated (beta-N-1-deoxy fructosyl) hemoglobin, and proinflammatory cytokines, and improved antioxidant and plasma lipid profiles in HFD/STZ-induced diabetic rats fed with 100 mg/kg rutin in their diet [97,98]. Particularly, rutin can meliorate metabolic abnormalities, oxidative stress, inflammation, and cellular apoptosis pathways in an STZ-induced diabetic rat model [99]. Kaempferol is an important flavonoid in herbal foods, and is reported to effectively inhibit inflammation and ameliorate insulin resistance by the beneficial regulation of the IRS-1 function [100]. Kaempferol exhibits anti-diabetic effect in multiple mechanisms, including anti-oxidative, anti-inflammatory, antihyperlipidemic, and pancreatic β-cell protection [101,102,103]. Recent studies show that HFD/STZ-induced diabetic rats ameliorate blood lipids and insulin with the treatment of kaempferol orally ingested for 10 weeks, and effectively restored IR induced alteration of glucose disposal through the inhibition of the phosphorylation of IRS-1, IKK/NF-κB signal and further proinflammatory cytokines tumor necrosis factor-α (TNF-α) and interleukin-6 (IL-6) levels [104,105]. Troxerutin is a flavonol, a type of flavonoid, derived from rutin, which can be used to treat thrombosis, cerebrovascular diseases, and edemas. However, troxerutin administration significantly reduced heart rate, blood pressure, blood glucose, and plasma triglyceride levels, as well as significantly reduced reactive oxygen species, NF-κB levels, and suppressed the phosphorylated forms of AKT, c-Jun N-terminal kinase (JNK) and IRS-1 in a rat model of T2D [106].

### 3.2. Isoflavone

Isoflavones are reported to have beneficial antioxidant and estrogenic effects in the treatment of cardiovascular diseases and may lower the risk of several cancers [107,108]. Isoflavones are predominantly found in soy beans and other leguminous plants, and are known to be the richest sources of the active isoflavones, including genistein and daidzein [109]. Genistein has the ability to increase insulin secretion in the pancreatic islets of adult mice through a reduced intracellular calcium concentration effect [110]. Genistein also induces cAMP, stimulates protein kinase A in pancreatic islet’s cell linings (PC12) [111], and regulates the insulin-induced glucose passage because of the conformational changes of the GLUT4, without affecting the translocation of GLUT4 in adipocytes [112]. Further studies showed that genistein acts as a direct inhibitor of the insulin-induced glucose passage in 3T3-L1 adipocytes [113]. In vivo experiments showed that dietary intake of genistein (250 mg·kg^−1^ diet) improved hyperglycemia, glucose tolerance, and blood insulin level, and increased the number of insulin-positive β-cell in islets, promoted islet β-cell survival, and preserved islet mass in HFD/STZ-induced diabetic mice [114]. In the ZDF rat model, genistein showed an increased GLUT4 expression level and larger soleus muscle fiber areas as well as regulation of estrogen receptor signaling pathways [115]. In the ovariectomized diabetic rat model, genistein has shown anti-diabetic and anti-inflammatory effects to markedly increase SIRT1 protein, and decreased IL-1β and NF-κB proteins levels compared to control groups [116]. SIRT1, a deacetylase, is expressed in the endocrine cells of the Langerhans islets and has been reported to inhibit NF-κB by deacetylating p65 and protect β-cells from cytokines [117]. In Lep^ob/ob^ mouse model, the genistein-fed group increases plasma levels of triiodothyronine (T3) and decreases in the protein expression of renal 11β-hydroxysteroid dehydrogenase type 2 (11β-HSD2), suggesting a thermogenic effect on energy expenditure, T3 production, and corticosterone status [118]. Daidzein belongs to the isoflavone subclass of flavonoids, and is present predominantly in the form of glucosides in various plants, including red clover, alfalfa, soybean, and some legumes [119]. Previous studies suggest that daidzein exerts anti-diabetic effects by improving glucose and lipid metabolism, and downregulating blood glucose, total cholesterol levels, and improving glucose uptake via GLUT4 translocation, as well as AMP-activated protein kinase (AMPK) activation [120,121,122,123]. In in vivo studies, Cheong et al. [124] demonstrated that 0.1% daidzein in the diet for 4 weeks suppressed the rise in fasting glucose, lipid levels, and IR and AMPK activation in gastrocnemius muscle compared to the diabetic (Lepr^db/db^ mouse) control group. In addition, daidzein also reduced blood glucose and urinary glucose excretion in KK-A^y^ mice [124]. Using Lepr^db/db^ mouse, another study by Park et al. [120] also demonstrated that daidzein (0.2 g/kg diet, 6 weeks) supplementation improved glucose and lipid metabolism and regulated hepatic glucose (GK, G6Pase, and PEPCK) and lipid- (FAS, CPT, and β-oxidation) regulating enzyme activities compared to those seen in diabetic control group.

### 3.3. Flavanones

Naringenin and hesperidin, the two major flavanones that are abundant in citrus fruits (such as grapefruit, orange and tomato) have been reported to possess antioxidant, anti-diabetic, lipid-lowering, anti-atherogenic, and anti-inflammatory activities [125,126]. Oral treatment of naringenin (25 mg/kg) exerts significant inhibition of intestinal α-glucosidase activity in vivo [127], thereby delaying the absorption of carbohydrates in HFD/STZ-induced diabetic rats, thus resulting in significant lowering of postprandial blood glucose levels. In the livers of Lepr^db/db^ mouse, naringin improves hyperlipidemia and hyperglycemia by partly regulating the fatty acid and cholesterol metabolism, and affecting the gene expression of glucose-regulating enzymes. Furthermore, naringin may upregulate hepatic and adipocyte PPARγ and GLUT4 to regulate the expression of hepatic enzymes involved in glycolysis and gluconeogenesis, thereby improving hyperglycemia [128]. Oral treatment of naringenin in the HFD/STZ-induced diabetic rat model involves enhancement of reverse cholesterol transport and paraoxonase activity [129], and decreases levels of glucose, HbA1c, MDA, NO, TNF-α, and IL-6 in HFD/STZ-induced diabetic rats [130]. Neohesperidin, a flavanone glycoside found in citrus fruits, is the 7-O-neohesperidose derivative of hesperetin. Neohesperidin significantly decreases serum triglycerides, total cholesterol and liver index, inhibits lipid accumulation in the liver, and decreases the size of epididymal adipocyte in the KK-A^y^ mice. Gene expression of stearoyl-CoA desaturase 1 (SCD-1) and fatty acid synthase (FAS) are significantly inhibited, and elevated level of phosphorylation of hepatic AMPK is observed in neohesperidin-treated mice. Therefore, the activation of the AMPK pathway and regulation of its target genes, including SCD-1, FAS, and ACOX, may play important roles in the hypoglycemic and hypolipidemic effects [131]. Hesperidin and naringenin, were reported to be beneficial for lowering blood glucose levels by upregulating hepatic glucokinase, PPARγ, and adipocyte GLUT4 in the Lepr^db/db^ mouse model [128].

### 3.4. Flavan-3-ols (Flavanols)

Drinking green tea has long been found to have an anti-diabetes effect [132]. Kim et al. [133] found that green tea catechins can protect the pancreas from oxidative damage, and green tea extract can enhance GLUT4 expression, increase glucose tolerance, promote glucose uptake, and decrease oxidative stress in diabetic rats [134]. Jingqi et al. [135] found that catechins exact anti-IR effect, significantly decrease glucose levels and increase glucose tolerance in KK-A^y^ animals by reducing ROS content and JNK phosphorylation and promoting GLUT4 translocation.

### 3.5. Flavones

Flavones differ from other flavonoids in that they have a double bond between C2 and C3 in the flavonoid skeleton, there is no substitution at the C3 position, and they are oxidized at the C4 position. Flavones from plants are typically conjugated as 7-O-glycosides; they contain apigenin, such as passionflower and chamomile, and have been used as traditional medicines for hundreds of years to treat a variety of diseases. In T2D studies, apigenin was intragastrically administered at 50 or 100 mg/kg once a day for 6 weeks in the HFD/STZ-induced diabetic rat model to investigate the effect of diabetes mellitus. Compared with the diabetic control group, apigenin significantly decreased the levels of blood glucose, serum lipid, malonaldehyde, ICAM-1 and IR index, and improved impaired glucose tolerance [136]. Another report investigated the role of apigenin in controlling damaged vital tissues in HFD/STZ-induced diabetic rats. Enhanced GLUT4 translocation and downregulated CD38 expression by apigenin were observed. Apigenin was also found to control blood glucose level, along with the protection of vital organs eventually damaged during diabetes. Baicalein is a flavonoid originally isolated from the traditional herbal remedy known as Chinese skullcap, which exhibits strong free radical scavenging [137]. Using the HFD/STZ-induced diabetic mouse model, and the administered 250 or 500 mg baicalein/kg diet, Fu et al. [138] found that baicalein treatment significantly improved hyperglycemia, glucose tolerance, insulin levels, and preserved islet mass by inhibiting apoptosis.

### 3.6. Anthocyanidins

Anthocyanidins are another class of flavonoids widely distributed in the human diet in apples, berries, red grapes, eggplant, red cabbage and radishes. Dietary consumption of anthocyanins is high compared to other flavonoids, owing to their wide distribution in plant materials. Considerable attention has been given to anthocyanins because of their potential health benefits, including anti-inflammatory, antioxidant, anti-obesity, and anti-diabetic effects. Cyanidin and its glycosides belong to anthocyanins and have been demonstrated to inhibit intestinal α-glucosidase and pancreatic α-amylase in vitro [139]. Cyanidin-3-glucoside, one of the most prevalent anthocyanins existing in our diet, ameliorates hyperglycemia and insulin sensitivity via the activation of AMPK in KK-A^y^ mouse [140], and this activation is accompanied by the up-regulation of GLUT4 in skeletal muscle and the downregulation of gluconeogenesis in the liver. Guo et al. [141] also showed that dietary cyanidin-3-glucoside significantly lowers fasting glucose levels and markedly improves the insulin sensitivity, macrophage infiltration, and the mRNA levels of MCP-1, TNF-α and IL-6 in adipose tissue, and modulates the JNK/forkhead box O1 signaling pathway in liver and adipose tissues of the HFD and Lepr^db/db^ mouse model.

## 4. Conclusions

Diabetes mellitus is now a major global public health problem. The existing animal therapeutic approaches and models to treat diabetes mellitus and obesity cannot replace clinical human disease studies. However, animal models of targeted T2D diseases offer useful complementary models for selected scientific questions, such as the study food intake, nutrient portioning, body fat distribution, systemic glucose metabolism, and many other key aspects of metabolic health and disease. Flavonoids are a potential alternative treatment strategy for the development of effective and safe anti-obesity and anti-diabetes drugs. Currently, there are no recommendations regarding the optimal content of flavonoids and their subclasses in a diet. However, the results of the presented studies prove a potential beneficial role of flavonoids in the prevention and treatment of T2D. This review suggests the important role of flavonoids in enhancing insulin secretion, reducing apoptosis and promoting the proliferation of pancreatic β-cells, reducing IR, inflammation, and oxidative stress in muscle and promoting translocation of GLUT4 via PI3K/AKT and AMPK pathways. Choosing the right models and using appropriate animal models provide successful discovery and development of safer and more potent therapeutics with the potential to stop the obesity and T2D pandemics.

## Figures and Tables

**Figure 1 nutrients-11-00530-f001:**
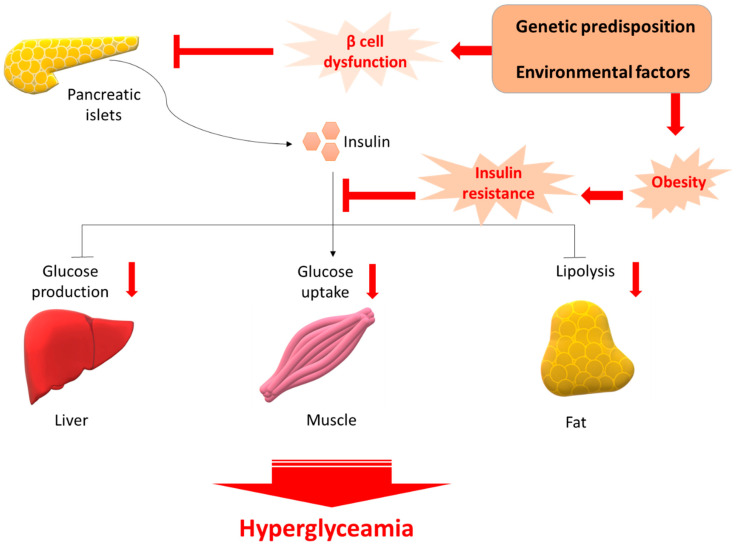
Pathology of T2D. β cell dysfunction and insulin resistance occurs following insult from several genetic predisposition and environmental factors. Initially the β-cell compensates by increasing the release of insulin; however, over time this compensatory mechanism fails and reduction in β-cell mass is evident. The reduced plasma insulin results in an increase in glucose levels. Glucose-sensitive tissues, including liver, muscle, and adipocytes, are unable to accommodate the increased glucose concentration. Persistent glucose release preserves the hyperglycemic environment, leading ultimately to T2D.

**Figure 2 nutrients-11-00530-f002:**
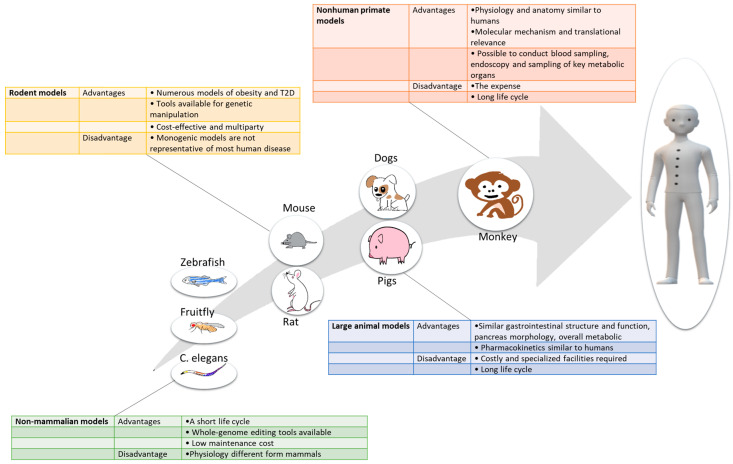
Major advantages and disadvantages of different classes of animal models used in T2D research.

**Table 1 nutrients-11-00530-t001:** List of selected rodent models potentially useful in type 2 diabetes research.

Strain or Method	Species	Obesity	Hyperpagia	Hyperglycaemia	Insulin Resistence	Hyperinsulinaemia	T2D
**Monogenic**							
**Lep^ob/ob^**	Mouse	++	+	+	++	++	-
**Lepr^db/db^**	Mouse	++	+	+	++	++	+
**ZDF**	Rat	-	-	+	++	+	+
**Polygenic**							
**C57BL/6J**	Mouse	+	+	-	+	-	-
**C57BL/6N**	Mouse	-	-	+	++	-	-
**KK-A^y^**	Mouse	+	+	+	++	+	+
**NZO**	Mouse	++	++	+	++	+	+
**TALLYO/Jng**	Mouse	+	+	+	++	+	+
**KK-A^y^**	Mouse	-	-	+	++	+	+
**Chemical-induced**							
**HFD/STZ**	Mouse, Rat	+	+	+	+	+	+

-, absent; +, mild; ++, severe; DIO, diet-induced obesity; HFD, high-fat diet. SD, standard diet; T2DM, type 2 diabetes mellitus; VMH, ventromedial hypothalamus.

**Table 2 nutrients-11-00530-t002:** Major subclasses of flavonoids with examples and some of the major dietary sources.

Flavonoid Subclasses	Color	Compounds	Dietary Sources
Flavanones 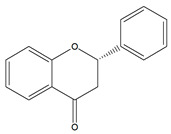	Colorless, Pale Yellow	Hesperetin, Naringenin, Eriodictyol, Naringin	Grapefruit, lemon, orange, grapefruit juice, lemon juice, orange juice.
Flavones 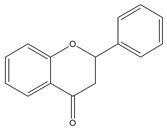	Pale Yellow	Luteolin, Apigenin, Vitexin, Orientin	Celery seed, dried parsley, thyme), celery, parsley, peppers.
Flavonols 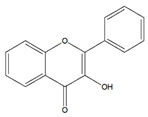	Pale Yellow	Quercetin, Kaempferol, Myricetin, Isorhamnetin, Rutin, Tiliroside, Aromadendrin, Silymarin, Silybin	Capers, apples, cranberries, arugula, asparagus, broccoli, cabbage, chives, coriander, endive, fennel, ginger, mustard greens, okra, onions, peppers, beans,
Flavan-3-ols (Flavanols) 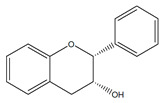	Colorless	Catechin, Gallocatechin, Epicatechin, Epigallocatechin, Epicatechin 3-gallate, Theaflavin, Theaflavin 3-gallate,	Apples, broad beans, pecans, pistachio, wine, cocoa, tea, soybeans.
Anthocyanidins 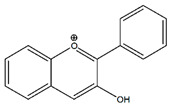	Blue, Red, Violet	Cyanidin, Delphinidin, Malvidin, Pelargonidin, Peonidin, Petunidin	Berries, blackberries, blueberry, cranberry, currants, grapes, plum, red cabbage, eggplant, pecans, pistachio, wine, black beans.
Isoflavones 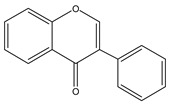	Colorless	Daidzein, Genistein, Glycitein	Red clover, soybeans and soybean products (milk, flour, yogurt and others).

**Table 3 nutrients-11-00530-t003:** Important anti-diabetic potential and the underlying mechanism of dietary flavonoids in selected rodent T2D models.

Flavonoids	Structural Formula	Sources	Pathways/Target Molecules	Experimental Model	References
**Flavonols**					
Quercetin		*Edgeworthia gardneri*	↑Intracellular Ca^2+^, ↑the ratio of Bcl-2/BAX, ↑mitochondria membrane potential.	Lepr^db/db^ mouse	[85,88]
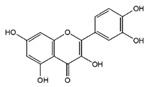			↑Serum insulin levels, ↓MIP-1α, Bax/Bcl2 ratio, protein disulfide isomerase activity	HFD/STZ-induced diabetic rat	[90]
Rutin		Buckwheat, oranges, grapes, lemons, limes, peaches and berries	↑Body weight, ↓plasma glucose and HbA1c, proinflammatory cytokines, ↑the depleted liver antioxidant status and serum lipid profile, ↑oxidative stress.	HFD/STZ-induced diabetic rats	[98,99]
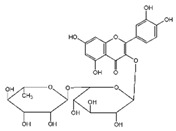			↑Insulin IRS-2/PI3K/Akt/GSK-3β signal pathway, ↑hepatocyte proliferation, ↓blood glucose level and generation of AGEs	Lepr^db/db^ mouse	[95]
Kaempferol 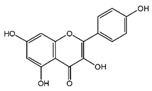		Tea, cruciferous vegetables, grapefruit, Gingko biloba L., and some edible berrie	↑Antioxidant status, ↓of lipid peroxidation markers, ↑membrane-bound ATPases.	HFD/STZ-induced diabetic rat	[103,105]
Troxerutin		Sophora japonica	↓Heart rate, blood pressure, blood glucose and plasma triglyceride levels	HFD/STZ induced diabetic rat	[106]
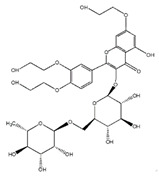					
**Flavanones**					
Naringin		Citrus fruits and Grapefrui	↓α-glucosidase activity	Lepr^db/db^ mouse	[125,127]
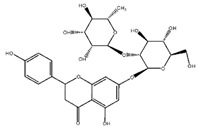			↑Cholesterol transport and paraoxonase activity,↓ levels of glucose, HbA1c, MDA, NO, TNF-α and IL-6	HFD/STZ-induced diabetic rats	[129,130]
Hesperidin		Citrus fruits	↓Hyperglycemia,↑plasma insulin, ↓PEPCK and G6Pase expression, ↑β-cell function	Lepr^db/db^ mouse	[128]
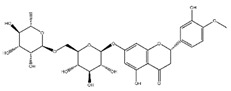					
**Isoflavones**					
Genistein		Soybeans and soy food	↓Plasma levels of corticosterone, ↓expression of 11b-HSD1	Lep^ob/ob^ mouse	[118]
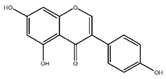			↓hyperglycemia, ↓glucose tolerance, and blood insulin level, ↑β-cell mass	HFD/STZ-induced diabetic rat	[114]
			↑GLUT4 expression level and larger soleus muscle fiber areas, ↑estrogen receptor signaling	ZDF rat	[115]
Daidzein		Soybeans and legumes	↓Fasting glucose, lipid levels and insulin resistance, ↑AMPK activation	KK-A^y^ and Lep^ob/ob^ mouse	[124]
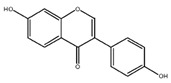			↑insulin/glucagon ratio, ↓hepatic glucokinase activity and hepatic fatty acid synthase, ↓plasma total cholesterol, triglyceride	Lepr^db/db^ mouse	[120]
**Anthocyanidins**					
Cyanidin 3-glucoside		Fruits, vegetables, berries, and red wine	↓ Blood glucose levels, ↑ insulinsensitivity, ↑AMPK activation, ↑GLUT-4,	KK-A^y^ mouse	[140]
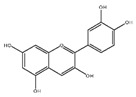			↓Macrophage infiltration and the mRNA levels of MCP-1, TNF-α and IL-6	HFD fed and Lepr^db/db^ mouse	[141]
**Flavan-3-ols (Flavanols)**					
Catechins 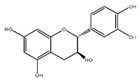		Green tea	↓Glucose levels,↑glucose tolerance,↓ROS decreased JNK phosphorylation,↑GLUT-4 translocation	KK-A^y^ mouse, HFD-induced obese rat	[135]
**Flavones**					
Apigenin 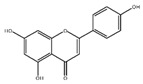		Celery, parsley	↓Glucose, serum lipid, ICAM-1 and insulin resistance index, ↑glucose tolerance	HFD/STZ-induced diabetic rat	[136]
Baicalein			↓Blood glucose, serum lipid, ↑SOD activity,↑glucose tolerance,	HFD/STZ-induced diabetic rat	[136]
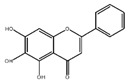			↓hyperglycemia, ↑insulin secretion	HFD/STZ-induced diabetic mouse	[138]

↑: increase; ↓: decrease.

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
