# Peer review of "In Vivo Rodent Models of Type 2 Diabetes and Their Usefulness for Evaluating Flavonoid Bioactivity"

_nutrients, 2019, doi:10.3390/nu11030530_

Round 1

Reviewer 1 Report

This review by Fang et al, discusses different models of T2D and their usefulness in studying the anti-diabetogenic effect of flavonoids. Overall, the review is comprehensive and well written, some clarifying points are of minor weakness:

1) Within each individual section of diabetes models - more focus should be spent on the advantages and disadvantages of each system.

2) The disadvantage of using monogenic forms of mice/rats as a model needs to be clearly stated.

3) The paragraph on polygenic models should be split into two separate paragraphs. One for polygenic models and one for diet-induced models.

4) Fig. 1 - the pancreas should also be labeled as islets.

4) Fig 2. The tables within fig 2 are confusing (specifically where the titles for advantage and disadvantage are). 

5) The text mentions table 3 - but there is no table presented within the manuscript.

Author Response

Reviewer #1:

Thank you for reviewing our manuscript and giving many useful comments to us. We had rewritten the manuscript according to the comments. I should like to express my appreciation to the referees for suggesting how to improve our paper.

1) Within each individual section of diabetes models - more focus should be spent on the advantages and disadvantages of each system.

Response 1: In this section, we had concluded at the end of each classification of diabetic animal models. And we also made Fig. 2 to let the reader quickly know the advantages and disadvantages of various diabetic animal models.

2) The disadvantage of using monogenic forms of mice/rats as a model needs to be clearly stated.

Response 2: We had pointed out the disadvantage of using monogenic model in Page 5, line 187-193.

3) The paragraph on polygenic models should be split into two separate paragraphs. One for polygenic models and one for diet-induced models.
Response 3: We had modified the content of polygenic models, divided it into two paragraphs, making it easier for readers to distinguish details.

4) Fig. 1 - the pancreas should also be labeled as islets.

Response 4: We had corrected the terms "pancreas" to "Pancreatic islets in Fig.1. Please check it in the revised version.

4) Fig 2. The tables within fig 2 are confusing (specifically where the titles for advantage and disadvantage are).

Response 4: We had modified the content of Fig.2 in Page 3, line 76-78. Please check it in the revised version.

5) The text mentions table 3 - but there is no table presented within the manuscript.

Response 4: We organized the table about "Important anti-diabetic potential and the underlying mechanism of dietary flavonoids in selected rodent T2D models" in table 3. We note it again. Please check it in the revised version.

Thank you for the valuable suggestions. We should like to thank the referees for their helpful comments and hope that we have now produced a more balance and better account of our work.

Reviewer 2 Report

This paper addresses an important topic, a role of flavonoids in diabetes treatment. The authors summarize recent progress in understanding the molecular mechanisms of flavonoids in animal models of type 2 diabetes as well as a review of currently available rodent animal models of type 2 diabetes. The reviewer thinks that this article is well written.

Author Response

Reviewer #2:

Thank you for reviewing our manuscript and giving many useful comments to us. We had rewritten the manuscript according to the comments. I should like to express my appreciation to the referees for suggesting how to improve our paper.